# Proliferation associated 2G4 is required for the ciliation of vertebrate motile cilia
Moonsup Lee[1], Christina Carpenter[2], Yoo-Seok Hwang[1], Jaeho Yoon [1], Quanlong Lu[3], Christopher J. Westlake[3], Sally A. Moody[4], Terry P. Yamaguchi [1] ✉ & Ira O. Daar [1] ✉

Motile cilia are critical structures that regulate early embryonic development and tissue homeostasis through synchronized ciliary motility. The formation of motile cilia is dependent on precisely controlled sequential processes including the generation, migration, and docking of centrioles/basal bodies as well as ciliary growth. Using the published proteomics data from various organisms, we identified proliferation-associated 2G4 as a novel regulator of ciliogenesis. Loss-of-function studies using *Xenopus laevis* as a model system reveal that Pa2G4 is essential for proper ciliogenesis and synchronized movement of cilia in multiciliated cells (MCCs) and the gastrocoel roof plate (GRP). Pa2G4 morphant MCCs exhibit defective basal body docking to the surface as a result of compromised Rac1 activity, apical actin network formation, and immature distal appendage generation. Interestingly, the regions that include the RNA-binding domain and the C-terminus of Pa2G4 are necessary for ciliogenesis in both MCCs and GRP cells. Our findings may provide insights into motile cilia-related genetic diseases such as Primary Ciliary Dyskinesia.

Motile cilia are microtubule-based complex hair-like structures that play a crucial role in embryo development and homeostasis. Mutations that cause defective multiciliation and abnormal cilia function lead to a variety of diseases known as ciliopathies[1–4]. In vertebrates, up to three hundred cilia are generated in multiciliated cells (MCCs) which are localized in specific tissues such as the spinal cord and brain ventricles, the oviduct and fallopian tubes, and the airways in mammals, and the epidermis in amphibians[5]. Mono-motile cilia located in the node in mouse embryos, the gastrocoel roof plate (GRP) in *Xenopus* embryos, and Kupffer's vesicle in zebrafish embryos generate a leftward flow of extracellular fluid that leads to the left-right asymmetry of organs during development[6–8].

Ciliogenesis in MCCs is a complex multi-step process[4]. During MCC differentiation, centrioles are amplified through deuterosome-dependent and centrosomal centriole-dependent pathways, leading to the formation of new procentrioles. These procentrioles grow and mature on deuterosome and centrosome platforms. Once matured, centrioles disengage from the platforms and undergo apical migration and docking to the cortical actin network in the membrane. The modified mature centrioles (basal bodies) nucleate motile cilia. The basal bodies form distal and subdistal appendages, and distal appendages are critical for basal body docking to the plasma membrane[9,10]. In mouse trachea MCCs, basal body docking to the apical membrane and ciliary vesicle formation is facilitated by the coordination of

Chibby, CEP164, Rabin8, and Rab8[11]. The basal body projects the basal feet towards the direction of ciliary beating, and the ciliary adhesion complexes are generated at the basal feet[12], and the ciliary adhesion complexes connect the basal bodies to the actin cytoskeleton[13,14]. In addition, microridge-like Ezrin actin anchoring complexes contribute to the basal body root-let mooring to the actin meshwork in MCCs[15].

Multiple cilia generate directional fluid flow through synchronized beating. The synchronized movement of cilia relies on a polarized array of basal bodies, which is regulated by hydrodynamic forces and cytoskeletal dynamics in conjunction with planar cell polarity signaling[16]. Furthermore, the unique structure of the motile cilium in MCCs, comprising nine microtubule doublets with two central microtubules in the axoneme, facilitates directional beating[17,18]. Cytoskeletal networks play essential roles in the synchronized beating of cilia in MCCs. Although it is known that actin negatively regulates primary ciliogenesis[19], the actin cytoskeleton networks contribute to basal body spacing and synchronized cilia beating in MCCs[20]. The apical migration and docking of newly formed centrioles to the surface depend on the actin cytoskeleton in the larval epidermis[21] and in the quail oviduct[22]. The actin cytoskeletal networks are modulated by small GTPases, which contribute to ciliogenesis. For example, RhoA is required for proper basal body docking and subsequent ciliogenesis in the murine airway epithelial cells and MCCs of *Xenopus* larval epidermis[21,23]. Likewise, the Rac1

[1]Cancer & Developmental Biology Laboratory, Center for Cancer Research, National Cancer Institute, National Institutes of Health, Frederick, MD, USA. [2]Electron Microscopy Laboratory, Frederick National laboratory for Cancer Research, Frederick, MD, USA. [3]Laboratory of Cell and Developmental Signaling, Center for Cancer Research, National Cancer Institute, National Institutes of Health, Frederick, MD, USA. [4]Department of Anatomy and Cell Biology, George Washington University, School of Medicine and Health Sciences, Washington, USA. ✉e-mail: yamagute@mail.nih.gov; daari@mail.nih.gov

regulator ELMO is required for multiciliation in *Xenopus* and zebrafish[24]. However, the role of actin cytoskeleton in mono-motile ciliogenesis is unclear.

To identify novel ciliary components and better understand their role in ciliogenesis, multiple proteomic studies have been conducted using various cells and tissues from different species[25–30]. We mined these studies[27,28,30] to identify proliferation-associated 2G4 (Pa2G4) as a potential ciliary component and/or ciliogenesis regulator. Pa2G4 is an evolutionarily conserved, multifunctional protein with two splice variants, p42 and p48. While the roles of Pa2G4 in cancer and other diseases have been extensively studied[30,31], few reports have addressed its role in embryonic development. Pa2G4 mutant mice exhibited defective organogenesis due to severe hemorrhage along with deregulated apoptosis and proliferation, and possibly through the gene-silencing protein DNMT1[32]. During amphibian development, Pa2G4 is required for neural crest and otic development in cooperation with Six1[33].

Here, we demonstrate for the first time a novel function of Pa2G4 in ciliogenesis during embryonic development. As suggested in the previous proteomic studies, Pa2G4 is observed in the cilia of *Xenopus* MCCs. Interestingly, the knockdown of Pa2G4 led to impaired cilium beating, likely due to a combination of abnormal cilium structure, apical actin malformation, disrupted basal body polarity and docking, and defects in distal appendage formation.

## Results

### Pa2G4 is required for ciliogenesis in MCCs

The previously published proteome data using different species[27,28,30], along with the localization of Pa2G4 transcripts in *Xenopus* larval epidermis[33,34], indicated a potential role of Pa2G4 in ciliogenesis and cilium beating. To assess the potential localization of Pa2G4 to the cilium and MCCs, we examined the expression of exogenous GFP-tagged Pa2G4 in *Xenopus* larval epidermis. 3D-Structured illumination microscopy (SIM) revealed numerous specks of GFP-Pa2G4 signal along axonemes (visualized with acetylated tubulin) and ciliary membrane (marked by membrane-RFP) (Fig. 1a). Interestingly, we observed strong GFP-Pa2G4 signals on the surface of MCCs. GFP-Pa2G4 partially overlapped with or was juxtaposed to the phalloidin-stained apical actin meshwork and was largely excluded from the distal appendage area, marked by Cep164 staining (Fig. 1b).

The localization of Pa2G4 to the axoneme and surface actin area of MCCs suggests a potential role in multiciliation and the function of motile cilia such as synchronized beating. This hypothesis was tested through loss-of-function analyses using antisense morpholino oligonucleotides (MOs) to inhibit endogenous Pa2G4 expression[33]. MOs, along with a membrane GFP mRNA cocktail, were microinjected into one ventral blastomere at the four- or eight-cell stage, the major progenitors of the epidermis[35]. Knockdown of Pa2G4 resulted in a dramatic, dose-dependent reduction in acetylated tubulin signal in MCCs (Fig. 1c, d). The Pa2G4 MO specificity was validated by co-injecting the MOs with either wild-type or rescue mRNA containing a modified MO binding site, followed by Western blot analysis (Supplementary Fig. 1a).

### Pa2G4 is required for cilia beating in MCCs

The compromised multiciliation on Pa2G4 morphant MCCs was quantified by measuring the cilia length and centriole number per MCC using scanning electron microscopy (SEM) and immunostaining, respectively. The average cilium length in Pa2G4 morphants (8.5 µm) was slightly decreased compared to the control (10.3 µm), and the number of centrioles per MCC was also reduced upon Pa2G4 knockdown (average centriole number per MCC: control (137), Pa2G4 morphant (106), rescue (121)) (Fig. 2a–c). The decrease in cilium length and centriole number was partially reversed by expressing wild-type Pa2G4, indicating that the phenotypes were due to reduced levels of Pa2G4.

Other populations of cilia observed in *Xenopus* embryos include the mono-motile cilia in the gastrocoel roof plate (GRP) region during the neurula stage and primary cilia in the neural tube. Since Pa2G4 is widely expressed in the roof of the gastrocoel (Supplementary Fig. 1b), we first assessed whether Pa2G4 is required for the ciliation of the GRP. In Pa2G4 morphants, not only was the length of GRP cilia shortened, but the cilia in GRP cells also lacked the typical posterior polarization observed in the control morphants (Supplementary Fig. 1c, d). Similarly, the neural tube of Pa2G4 morphants displayed a shortened cilia population compared to the control side (Supplementary Fig. 1e).

Because the shortened cilia in the GRP of Pa2G4 morphants could lead to a loss of leftward flow in the GRP and disrupt asymmetric embryonic development, we tested the asymmetric expression of the Nodal inhibitor *dand5* (Coco), along with the Nodal downstream gene *pitx2c*, which serve as direct indicators of the leftward flow[36,37]. Pa2G4 knockdown decreased right-biased Coco expression to 35% from 70% in controls, while Pa2G4 mRNA expression restored it to 59% (Supplementary Fig. 2a). Additionally, the Pa2G4 morphants lost asymmetrical *pitx2c* expression on the left side (Supplementary Fig. 2b), suggesting that the leftward flow may be disturbed by the shortened GRP cilia. These findings indicate that Pa2G4 is required for ciliogenesis in both MCC and GRP cells.

Multiciliated cells generate directional fluid flow through synchronized ciliary beating which is essential for proper development and tissue homeostasis[1,4]. To determine if Pa2G4 is necessary for ciliary beating in MCCs, high-speed imaging was performed. Compared with control morphant MCCs, Pa2G4 morphant MCCs exhibited defective ciliary beating phenotypes, ranging from completely immotile cilia to slow and irregular beating cilia (Supplementary movies 1–3). Kymographs revealed significant cilia-beating defects in Pa2G4 morphant MCCs and the impaired movement was restored by wild-type Pa2G4 expression (Fig. 2d). Interestingly, transmission electron microscopy (TEM) showed the formation of microtubule structure components in the axoneme were defective, with an abnormal number of outer microtubule doublets and missing central pairs (e.g., 8 + 0 and 9 + 0 rather than 9 + 2) (Fig. 2e). Given that a central pair of microtubules in ciliary axonemes is necessary for proper cilia beating[38,39], the inaccurate microtubule arrangement and absence of the central pair might result in cilia beating defects in Pa2G4 morphants.

Furthermore, previous reports describing the role of polarized basal body orientation in MCCs in coordinating ciliary beating[40,41] led us to test the planar polarity of basal bodies in MCCs by co-expressing CLAMP-GFP as a rootlet marker and centrin-RFP as a basal body marker. Compared with the control MCCs, the orientation of centrin-RFP and CLAMP-GFP was disorganized in Pa2G4 morphant MCCs (Fig. 2f). Thus, the ciliary beating defects in Pa2G4 morphant MCCs are due to deregulated ciliogenesis as well as randomized cilia orientation.

### Pa2G4 plays a role in apical actin formation by modulating Rac1 activity in MCCs

Although short cilium length is a common consequence of mutations that affect the central pair[39,42], ciliogenesis is influenced by several other factors including actin meshwork formation, and migration and docking of basal bodies to the surface[19,43,44]. To assess whether the apical actin meshwork is properly formed in Pa2G4 morphant MCCs, the apical actin structure was visualized using phalloidin staining. The morphant MCCs exhibited decreased acetylated tubulin staining with a significant reduction in phalloidin signal, and wild-type Pa2G4 expression was sufficient to restore the diminished apical actin meshwork (Fig. 3a), suggesting that Pa2G4 is necessary for apical actin meshwork formation. Concomitantly, Pa2G4 morphant MCCs displayed abnormal behavior of Centrin-RFP with defective migration and docking to the surface (Fig. 3b), consistent with previous reports on the requirement of the actin meshwork for basal body migration and docking[19].

The active small GTPases such as RhoA and Rac1 contribute to ciliogenesis in part by regulating basal body docking[21,23,24]. Fluorescent protein-tagged rGBD, pGBD, and wGBD can be used to monitor the spatiotemporal activity of these GTPases. The GTPase binding domains (GBDs) of effector proteins bind only to the active conformation of Rho GTPases. Thus, active Rho, Rac, and Cdc42 are detected by rGBD (GBD of

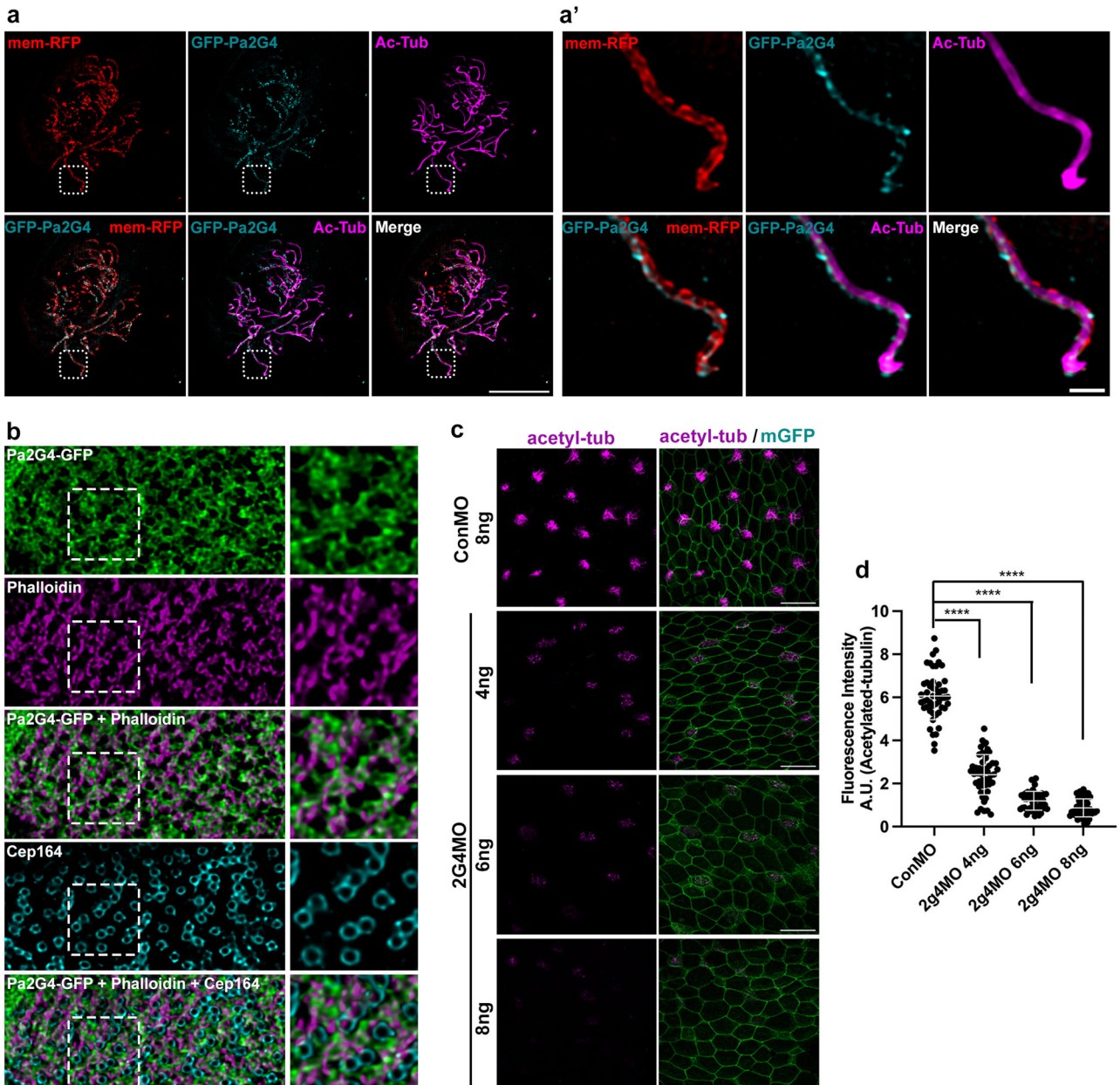

**Fig. 1 | Pa2G4 is required for ciliogenesis in MCCs. a, a'** 3D-SIM volume maximum intensity projection of multi-cilia in a MCC. Pa2G4-GFP (cyan) and mem-RFP (red) mRNAs were microinjected into a ventral blastomere at the four-cell stage. Pa2G4-GFP was observed in MCC cilia. Cilia were visualized by acetylated tubulin staining. White squares were magnified to the panel (**a'**). scale bars, 10 μm (**a**) and 2 μm (**a'**). **b** 3D-SIM volume maximum intensity projection of a MCC. Pa2G4 localized near or on the apical actin structure. Apical actin was shown by phalloidin staining (magenta), and endogenous Cep164 was shown by Cep164 antibody staining (cyan). The embryos were observed at stage 30. Scale bar, 2 μm. **c** Pa2G4 knockdown causes deceased acetylated tubulin signal in MCCs in a MO dose-dependent manner. The indicated amount of MOs with membrane-GFP mRNA (as a tracer) were injected into both ventral blastomeres at 4-cell stage embryo. Multi-cilia (magenta) were visualized by immunofluorescence by anti-acetylated tubulin antibody staining. **d** Relative acetylated tubulin signal intensity is quantified from c (image n = 50 from 25 embryos per group). ****, P < 0.0001, one-way ANOVA; scale bars, 50 μm. Error bars indicate ± SD.

Rhotekin), pGBD (GBD of PAK3), and wGBD (GBD of N-WASP), respectively. To assess whether Pa2G4 knockdown affects the activity of Rho family GTPases in MCCs, GFP-tagged rGBD, pGBD, and wGBD mRNAs were co-injected with centrin-RFP mRNAs and MOs. In control MCCs, all rGBD-GFP (Rho marker), pGBD-GFP (Rac1 marker), and wGBD-GFP (Cdc42 marker) were enriched near centrin-RFP (basal body marker), suggesting that active Rho, Rac1, and Cdc42 accumulate in the basal body area (Fig. 3c, d and Supplementary Fig. 3). While the localization of rGBD-GFP and wGBD-GFP to basal bodies was not disturbed by Pa2G4 knockdown, Pa2G4 morphant MCCs showed a significant reduction of pGBD-GFP signal at the basal bodies (Fig. 3c, d and Supplementary Fig. 3).

However, the basal body localization of GFP-Rac1 remained unchanged by Pa2G4 knockdown (Fig. 3e, f), indicating that Pa2G4 regulates Rac1 activity without affecting its localization. Since the basal body signal from pGBD-GFP at GRP cilia was diminished upon Pa2G4 knockdown (Supplementary Fig. 4), the regulation of Rac1 activity by Pa2G4 is not limited to the MCCs but might occur in other tissues such as the GRP.

**Pa2G4 is required for distal appendage formation**

Cep164, a 164kDa centrosomal protein, is an essential component of distal appendages and regulates basal body docking during the ciliogenesis of both primary cilia and multi-cilia[45–47]. To test whether the distal appendages are

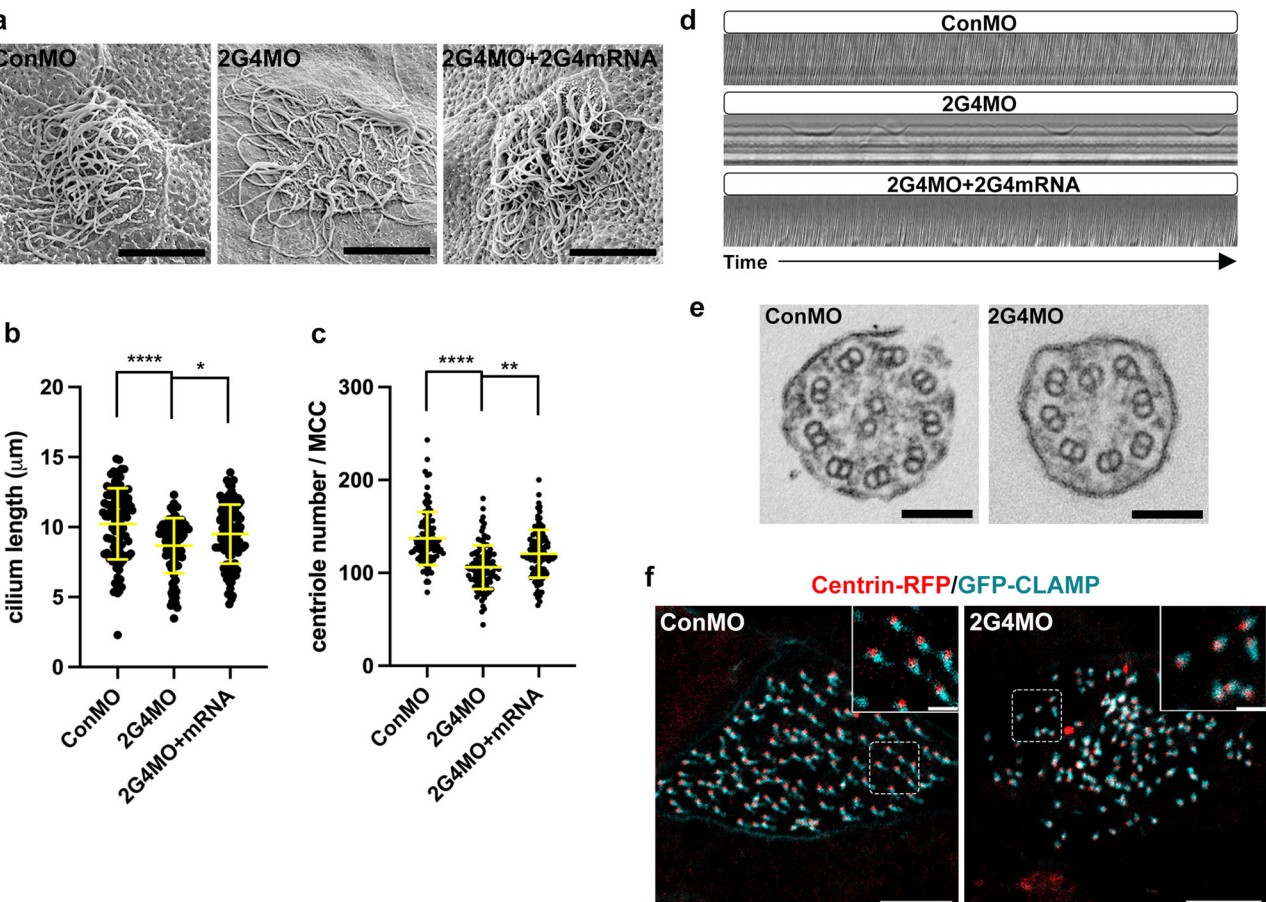

**Fig. 2 | Pa2G4 is required for cilia beating in MCCs. a** Pa2G4 knockdown phenotypes are confirmed with Scanning EM. A cocktail of Pa2G4 MO (total 8 ng, 4 ng of MO1, and 4 ng of MO2) with or without rescue Pa2G4 mRNA was injected into both ventral blastomeres at the four cell stage, and embryos were fixed at stage 32. Scale bars, 2.5 μm. (b and c) length of cilia and population of centrioles in MCCs decrease upon Pa2G4 knockdown. **b** Ciliary length quantification. Measured cilia number n > 100; embryos per group, n = 4; ****, p < 0.0001; *, P = 0.0229; error bars represent SD. **c** Centriole number (per MCC) quantification, MCC n = 106; embryos per group n = 15; ****, p < 0.0001; **, p = 0.002; error bars represent SD. **d** Defective ciliary beating in Pa2G4 morphant MCCs. Kymographs of ciliary beating of conMO,

Pa2G4MO, and rescue morphant MCCs. The X-axis of images represents time (5 seconds). **e** Transmission EM of cilia in control and Pa2G4 morphant MCCs. The representative images show a missing central pair in cross-sectioned Pa2G4 morphant multi-cilia. Scale bars, 100 nm. **f** Planar polarization defect of basal body-rootlet in Pa2G4 morphant MCCs. Rootlets and basal bodies are visualized by GFP-CLAMP (cyan) and centrin-RFP (red), respectively. Maximum intensity projection from serial section z-stacks was applied to generate images. Areas enclosed by the white dotted square are magnified in the right upper corner insets. Scale bars, 5 μm, 1 μm (insets).

properly formed in Pa2G4 morphant MCCs, endogenous Cep164, and centrin were stained and then visualized using 3D-SIM. While control and rescued MCCs displayed donut-shaped structures (marked by Cep164) surrounding the basal bodies (marked by centrin), irregular structures formed near basal bodies in Pa2G4 morphant MCCs (Fig. 4a, b). This suggests that Pa2G4 may contribute to the formation of distal appendages.

### The RBD and C-terminal regions of Pa2G4 are required for multiciliation

The full-length Pa2G4 in *Xenopus laevis* embryos is equivalent to the human p48 isoform[33], and the amino acid sequence alignment shows *Xenopus* full-length Pa2G4 is almost identical to the human p48 isoform (Supplementary Fig. 5a). Based on a previous report[31], deletion mutants were generated to test whether deleted domains are necessary to rescue the knockdown phenotypes in ciliogenesis. Initially, the localization of these mutants in MCC cilia was assessed using acetylated tubulin staining. Intriguingly, the localization of ΔRBD and ΔC mutants to cilia was compromised, compared to wild-type and the ΔN mutant (Fig. 5a). Consistent with the localization data, the ΔRBD and ΔC failed to restore the knockdown phenotypes in MCCs, as evidenced by reduced phalloidin, acetylated tubulin staining, and basal body docking to the apical surface (Fig. 5b–g). Furthermore, while the expression

of Pa2G4 WT and ΔN rescued the decreased Rac1 activity, the expression of ΔRBD and ΔC failed to recover Rac1 activity in Pa2G4 morphant MCCs, suggesting that the RBD and C-terminal regions may be critical for Rac1 activation in MCCs (Fig. 5h, i).

Similarly, although wild-type and ΔN expression partially rescued the knockdown-mediated shortened cilia in the GRP, neither ΔRBD nor ΔC expression was sufficient to rescue the knockdown phenotypes in GRP (Supplementary Fig. 5b, c).

## Discussion

The application of proteomic approaches across various model systems has led to the identification of numerous potential ciliary components and signaling proteins. However, only a limited portion of these proteomic data has been validated, suggesting that additional potential therapeutic targets for ciliopathies await discovery. From the proteomic data pools, Pa2G4 was identified as a potential ciliary protein. Although the role of Pa2G4 in cancer and mammalian cell model systems has been extensively studied, a potential function for Pa2G4 in ciliogenesis remains unknown. In this report, we demonstrate that Pa2G4 is required for the formation and function of both mono-motile cilia (GRP cilia) and multi-cilia (MCC cilia) in *Xenopus*.

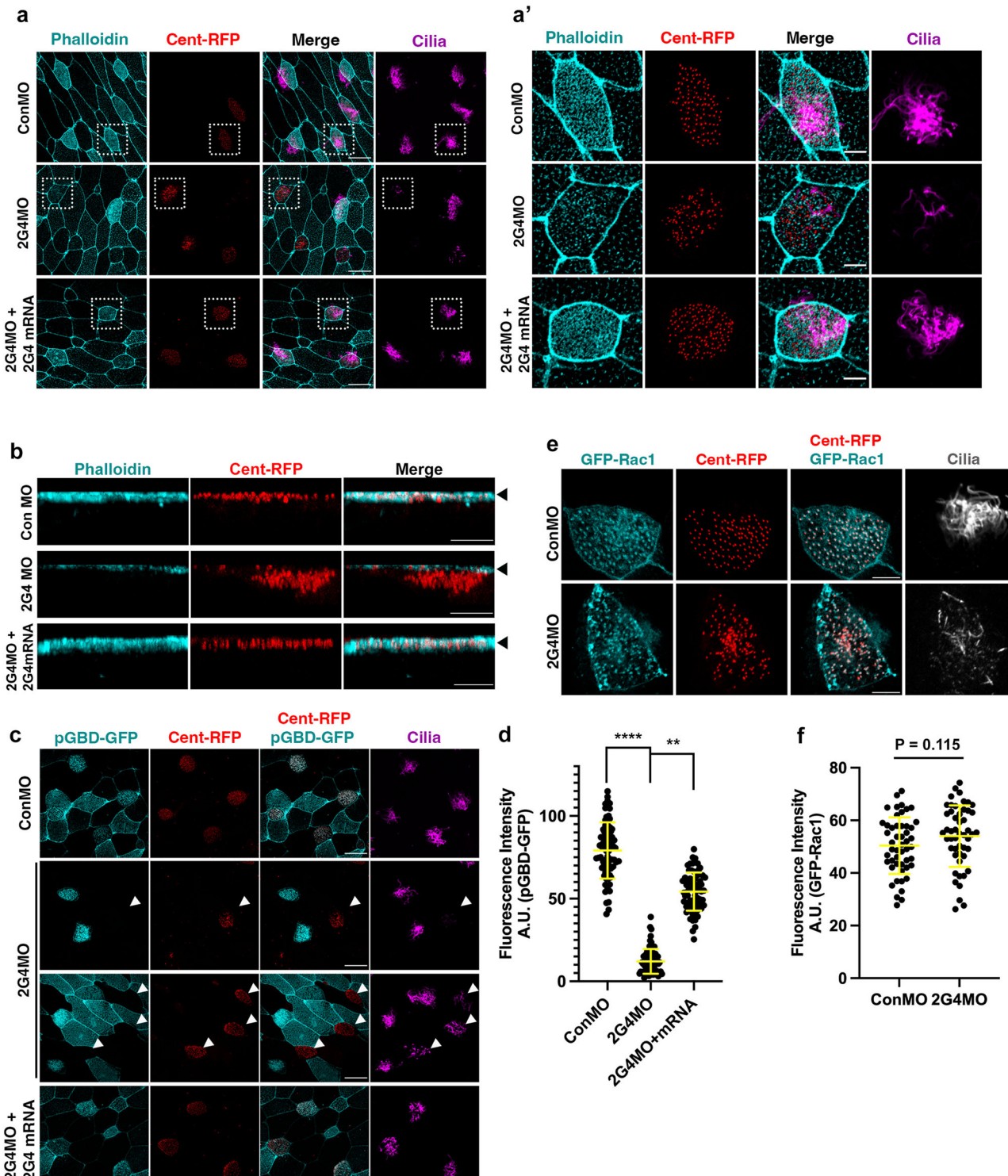

**Fig. 3 | Pa2G4 plays a role in apical actin formation by modulating Rac1 activity in MCCs. a, a'** Apical actin signal decreased in Pa2G4 morphant MCCs. MOs and mRNAs of centrin-RFP and wild-type Pa2G4 were injected into one ventral blastomere at the eight-cell stage. Phalloidin (cyan) was used for apical actin staining. White squares were magnified at (**a'**). Scale bars, 20 μm (**a**) and 5 μm (**a'**). **b** Defective basal body docking at Pa2G4 morphant MCCs. Centrin-RFP mRNA was coinjected with the indicated MOs and mRNAs. Z-stack confocal images were projected in the x-z plane. MCCs apical surfaces marked by arrowheads. Scale bars, 5 μm. **c** Pa2G4 knockdown decreased pGBD-GFP signal in MCCs. pGBD-GFP mRNA was injected into two ventral blastomeres in four-cell stage embryos, followed by injecting with

Centrin-RFP and MO (6 ng) with or without Pa2G4 WT mRNA into one ventral blastomere at the eight-cell stage. Anti-acetylated tubulin staining represents multi-cilia (magenta). White arrow heads indicate the MCCs expressing Centrin-RFP with Pa2G4 MO. Scale bars, 20 μm. **d** Quantification of pGBD-GFP intensity from (**c**), MCCs n = 73; embryos per group n = 18; one-way ANOVA, ****, p < 0.0001; error bars represent SD. **e** Pa2G4 knockdown does not change localization pattern of GFP-Rac1 in MCCs. Multi-cilia (gray) were stained with anti-acetylated tubulin antibodies. Scale bars, 5 μm. **f** Quantification of GFP-Rac1 intensity from (**e**), MCC n = 50; embryos per group n = 15; unpaired two-tailed Student's t-test, error bars represent SD.

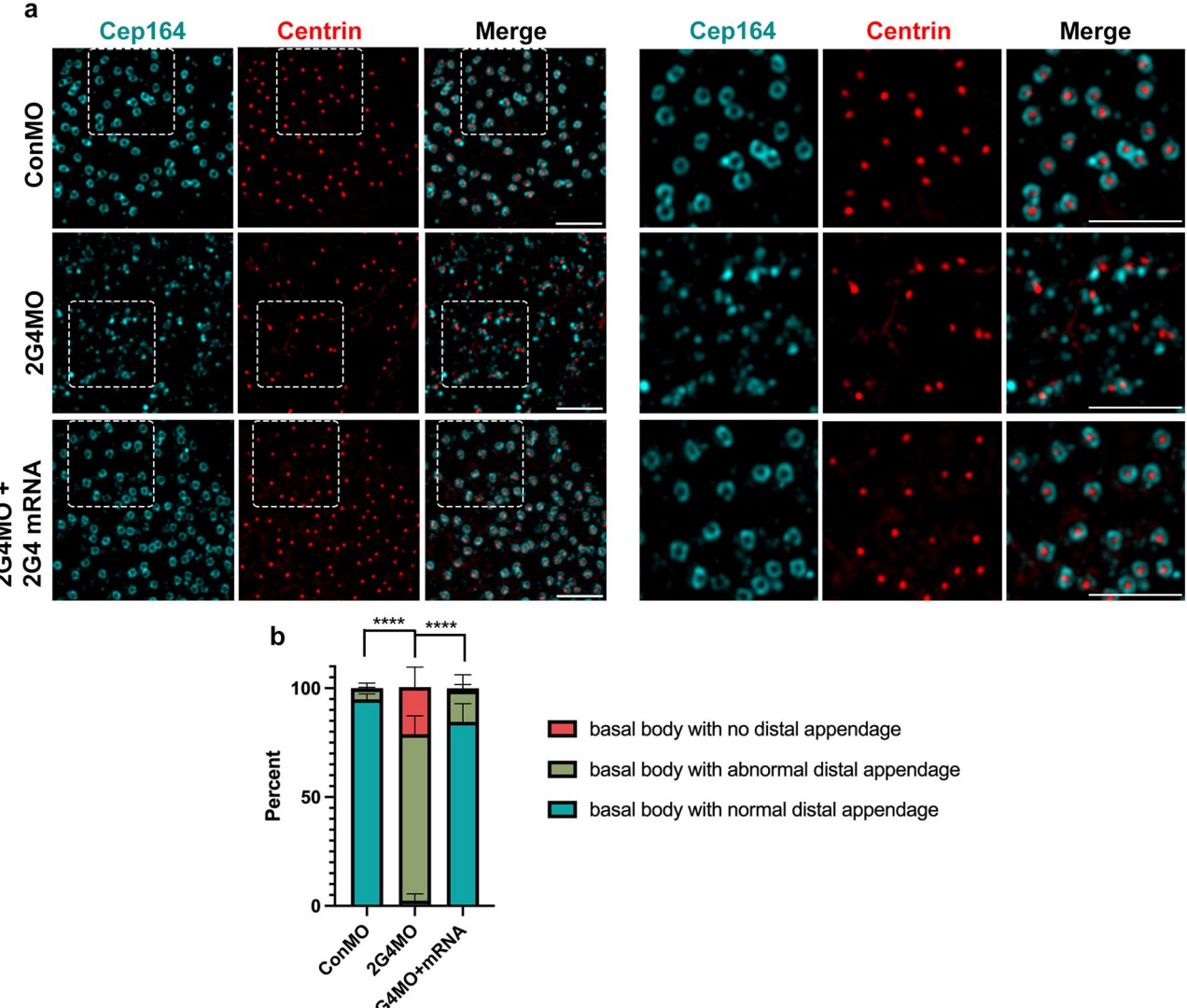

**Fig. 4 | Pa2G4 is required for distal appendage formation. a** 3D-SIM volume maximum intensity projection of MCC apical surfaces. A cocktail of the indicated mRNAs and MOs was injected into one ventral blastomere of eight-cell stage embryos, and embryos were harvested at stage 28. Endogenous Cep164 (cyan) and centrin (red) were visualized by immunostaining. White squares were magnified to the right. Scale bars, 2.5 μm. **b** Quantification of basal body and distal appendage staining images. The number of basal bodies with distal appendages per image were manually counted based on distal appendage shapes. (definitions—basal body with normal distal appendage: a basal body with a complete donut-shaped distal appendage, basal body with abnormal distal appendage: a basal body with an incomplete or irregularly-shaped distal appendage, basal body with no distal appendage: centrin (red) signal alone), Image n = 40; embryos per group n = 12; one-way ANOVA, ****, p < 0.0001; error bars represent SD.

Consistent with the localization of Pa2G4-GFP to cilia and the apical region of MCCs (Fig. 1a, b), Pa2G4 knockdown disrupted multiciliation and synchronized ciliary beating (Figs. 1c, d and 2). The actin cytoskeletal network is significantly correlated with ciliary formation and function in MCCs. Our findings that apical actin formation was compromised in Pa2G4 morphant MCCs (Figs. 3a, 5c) suggest that Pa2G4 might be required for actin cytoskeleton enrichment in MCCs. This is supported by previous reports in which Pa2G4 overexpression induced F-actin in hepatocellular carcinoma[48]. Furthermore, the reduced Rac1 activity upon Pa2G4 knockdown in MCCs (Figs. 3c, d, 5h, i, and Supplementary Fig. 3) suggests a potential role for Pa2G4 in remodeling the actin meshwork, in cooperation with Rac1 and its regulator ELMO.

Although Rho, one of the small GTPases is known for its role in ciliogenesis[21,23], Rac1 also contributes to ciliogenesis across different species. For instance, Rac1 regulates cilia length in pronephric tubules in zebrafish embryos and basal body docking and spacing in MCCs during *Xenopus* epidermis development[24]. Additionally, Rac1 is reported to increase IFT88 stability for proper ciliogenesis in the C3H10T1/2 cell line[49], and defects in Rac1 activity are associated with abnormal primary cilia assembly in Lowe syndrome[50]. Furthermore, inhibition of Rac1 impairs posterior basal body positioning and unidirectional fluid flow in the murine node[51], which is consistent with the centralized rather than posteriorly tilted position of cilia in Pa2G4 morphant GRP cells (Supplementary Fig. 1c, d, 4 and 5b, c). However, since Rac1 inhibition negligibly affects axonemal length in nodal cilia[51], the shortened cilia phenotype in the GRP observed with Pa2G4 knockdown may be independent of an ELMO-Rac1 module.

In addition to the importance of actin structure formation in ciliogenesis and ciliary beating in MCCs, deregulated basal body migration and docking also influence these processes in MCCs. Basal body docking largely depends on the ciliary vesicles attached to the distal appendages of the basal body, and Cep164, a distal appendage protein, plays a critical role in ciliary vesicle formation and basal body docking by recruiting other distal appendage proteins such as Chibby1 and FAM92[45,46]. Our finding that Pa2G4 knockdown compromises Cep164 recruitment to basal bodies

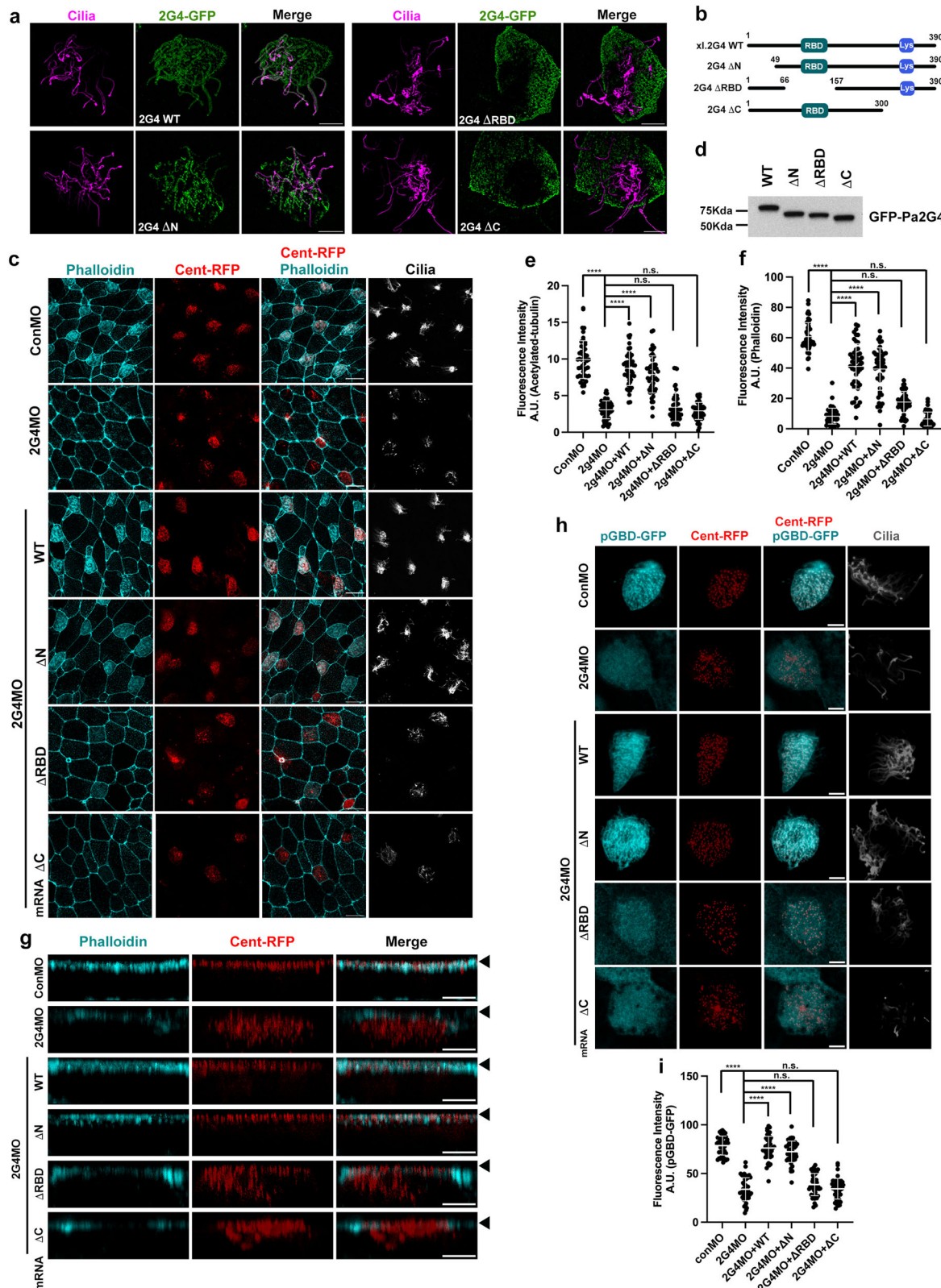

(Fig. 4) suggests that the recruitment of Chibby1 and FAM92 to the distal appendage and subsequent assembly of ciliary vesicles could be impaired, leading to basal body docking defects.

In addition to Pa2G4's role in basal body docking in MCCs, Pa2G4 also regulates axonemal structure. Most motile cilia and flagella contain $9 + 2$ microtubule structure, which includes nine outer doublets and a pair of

central microtubules. The pair of central microtubules form a central apparatus with hundreds of surrounding proteins that are crucial for ciliary beating[52,53]. Defects in the central apparatus, such as a missing central pair, are considered one of the diagnostic indicators of Primary Ciliary Dyskinesia (PCD), a genetic disorder of motile cilia[54]. Therefore, the absence of a central pair in the axoneme of Pa2G4 morphant MCC cilia (Fig. 2e) suggests

**Fig. 5 | The RBD and C-terminal regions of Pa2G4 are required for multi-ciliation. a** ΔRBD and ΔC mutants did not co-localize with the acetylated tubulin signal (magenta). The images were taken by 3D-SIM. Scale bars, 5 μm. **b** Protein domain schematic of *Xenopus* Pa2G4 wild-type and deletion mutants used for rescue experiments. RBD, RNA binding domain; Lys, Lysine rich region. **c** ΔRBD and ΔC mutants failed to rescue ciliary phenotypes upon Pa2G4 knockdown. The indicated mRNAs and MOs were injected into one ventral blastomere at the eight-cell stage. Phalloidin (cyan) stains apical actin and Centrin-RFP marks basal bodies. Cilia was visualized by acetylated tubulin staining. Scale bars, 20 μm. **d** Western blot of the indicated exogenous Pa2G4 proteins in (**c**). Quantification of acetylated tubulin signal (**e**) and phalloidin intensity (**f**) in an MCC shown in (**c**). ****, P < 0.0001, one-way ANOVA; image number for acetylated tubulin staining analysis, n = 50; embryos per group (**e**), n = 25; MCCs for phalloidin intensity quantification, n = 50; embryos per group (**f**), n = 10; Data are mean ± SD. The expression of ΔRBD and ΔC failed to recover basal body docking (**g**) and Rac1 activity (**h**, **i**) in Pa2G4 morphant MCCs. **g** The cocktail of the indicated MOs and mRNAs was microinjected into one ventral blastomere at the eight-cell stage. The representative z-stack confocal images were projected in the x-z plane. MCC apical surfaces marked by arrowheads. Scale bars, 5 μm. **h** pGBD-GFP and Centrin-RFP mRNAs were co-injected with the indicated Pa2G4 mRNAs and MOs (4 ng) into one ventral blastomere of four-cell stage embryos. Anti-acetylated tubulin staining represents multi-cilia (gray). Scale bars, 5 μm. **i** Quantification of pGBD-GFP intensity in (**h**), MCC n = 48; embryos per group n = 12; one-way ANOVA, ****, p < 0.0001; error bars represent SD.

that Pa2G4 regulates ciliary motility in MCCs by maintaining the central apparatus.

Pa2G4 participates in a broad range of cellular events including proliferation, differentiation, and apoptosis through interactions with proteins as well as RNA/DNA[55]. The C-terminal region of Pa2G4 includes an LXXLL motif, through which a variety of proteins (e.g., Rb, Akt, histone deacetylase 2, and androgen receptor) may interact with Pa2G4[31]. Additionally, a lysine-rich sequence ([364]RKTQKKKKKK[373]) in the C-terminal region and the putative double-strand RNA binding domain is important for Pa2G4 binding to RNA. Our data showed mutants with deletion of the regions that include the LXXLL and RNA binding domain failed to rescue Pa2G4 knockdown phenotypes, indicating the interaction with other proteins or RNAs is likely necessary for ciliogenesis (Fig. 5b-i). The potential binding proteins could be a component(s) of intraflagellar transport (IFT) machinery. Our data also showed that the ciliary localization of the ΔRBD and ΔC mutants was strongly compromised (Fig. 5a), suggesting that the regions containing the RBD and C-terminus are necessary for Pa2G4 translocation to a cilium. IFT trains transport ciliary proteins to the ciliary tip and contribute to ciliary assembly[56]. Assuming that Pa2G4 proteins are localized to the cilia by the IFT machinery, the deleted regions of Pa2G4 might be critical for interacting with IFT components.

Taken together, Pa2G4, a potential ciliary protein, regulates ciliogenesis and ciliary motility in both MCCs and GRPs by modulating apical actin enrichment, basal body docking, and distal appendage formation. Our study is expected to contribute to expanding knowledge on ciliary dynamics and function in motile cilia.

## Material and methods

### *Xenopus* embryo, microinjection, morpholino

Wild-type *Xenopus laevis* embryos were prepared for microinjection by the previously described method[57]. Briefly, eggs were collected from female frogs injected with human chorionic gonadotropin at 600U one day before microinjection, and the eggs were in vitro fertilized with sperm from dissected male testes. The capped mRNAs were synthesized with the SP6 mMessage mMachine Kit (AM1340). Two translation-blocking Pa2G4 morpholinos used for this study are described in the previous report[33] and were purchased from Gene Tools: Pa2G4_MO1, TACAGCCCA CTGCTTCTTGTCGTCCTC; Pa2G4_MO2, TCCCCACTTTCTCGAC AGTGTCC. An equimolar mixture of two Pa2G4 morpholinos (4–8 ng) was microinjected into one or two ventral blastomeres of the 4- or 8-cell stage embryos. The microinjected embryos were cultured at 18 °C until harvest.

All animal procedures were conducted by NCI-CCR affiliated staff and approved by the NCI Animal Care and Use Committee (ACUC) in compliance with federal regulatory standards, and AAALAC International accredits all components of the intramural NIH ACU program (animal study protocol # 23-433).

### Plasmids and cloning

HA-tagged *Xenopus laevis pa2g4* has been described[33]. GFP-tagged *pa2g4* was generated by PCR amplification and subcloned into pCS2 vector. The deletion mutant of *pa2g4* (ΔRBD) was generated by site-directed mutagenesis and ΔN and ΔC mutants were cloned by PCR amplification. Morpholino-resistant Pa2G4 cDNA was generated by mutating the MO binding region (from ATG TCG GGT GAC GAA GAA CAG CAG to ATG TCC GGA GAT GAG GAA CAA CAA). Full-length and mutants were linearized with NotI, and mRNAs were synthesized using the SP6 mMessage mMachine kit (Invitrogen). Centrin4-RFP and GFP-CLAMP constructs, gifts from Dr. Wallingford, were utilized for cilia polarity test. Coco (*dand5*: full-length CDS probe)[58] construct, a gift from Dr. A. H. Brivanlou, and *pitx2c* construct (348 bp of accession number AJ243596)[59], a gift from Dr. M. Blum were used for wholemount in situ assay. Pa2g4 anti- and sense-RNA probes (total 552 nucleotides = 184 nucleotides from 5′ UTR and 368 nucleotides from CDS) were generated from *Xenopus pa2g4* cDNA(NM_001090524.1). The sense and antisense DNA fragments were PCR-amplified and subcloned into pCS2 vector (Anti-sense: F primer_GATC CTCGAG GACGTTCAACGGTGTCTGTG and R primer_GATC GAATTC TGGGCAACATTAGCAATGAA; Sense: F primer_GATC GAATTC GACGTTCAACGGTGTCTGTG and R primer_GATC CTCGAG TGGGCAACATTAGCAATGAA). The following plasmids were purchased from Addgene: GFP-pGBD (#26735), GFP-wGBD (#26734), GFP-rGBD (#26732).

### Immunostaining and microscopy

Embryos were fixed with 1x MEM salt with 4% formaldehyde for 1 h at RT and dehydrated with methanol overnight at −20 °C. Dehydrated embryos were sequentially rehydrated. In case the dehydration step was omitted, embryos were permeabilized in PBS with 0.3% Triton for 30 min at RT. Blocking was conducted with whole-mount block solution (WMBS) for 1 hr at RT and primary antibody incubation was followed overnight at 4°C. The antibodies used for immunostaining are as follows: anti-acetylated tubulin (dilution 1:1000; T7451; Sigma-Aldrich), chicken anti-GFP (dilution 1:500; NB100-1614; Novus Biologicals), anti-Cep164 (dilution 1:250; 22227-1-AP; Protein Tech), and anti-centrin (dilution 1:250; 04-1624; EMD Millipore). Alexa Fluor 488 phalloidin (dilution 1:500; A12379; Thermo Fisher Scientific) and Alexa Fluor plus 405 Phalloidin (dilution 1:400; A30104; Thermo Fisher Scientific) were used for filamentous actin staining. Confocal microscopy was performed with an LSM 880 microscope equipped with a Plan-Apochromat 63 × /1.4 oil DIC M27 objective. Serial z-stack images were taken over 0.2 μm (for a single MCC) and 0.8 μm (for a multiple MCC).

For basal body migration docking assay, serial z-stack images were taken over 0.1 μm and total 5–6 μm distance and projected in the x-z plane.

3D structured SIM was conducted with N-SIM microscope (Nikon Instruments) equipped with a 100×/1.49 NA Plan Apo oil objective and an Andor iXON DU-897E camera. and Elyra7 with lattice SIM[2] (Zeiss) equipped with alpha Plan-Apochromat 63x/1.46 oil objective and two pco.edge sCMOS cameras. Serial z-stacks were taken over 0.1 μm distance in 3D-SIM mode, and raw images were reconstructed for generation of super-resolution images. 0.2 μm TetraSpeck microspheres (T7280; Thermo Fisher Scientific) were utilized to correct an xyz shift. Antibody-stained whole-mount embryos were imaged by identical settings for the same groups, and generated images were processed with identical conditions.

For cilia beat analysis, ciliary motility recordings at stage 32 embryos were collected using Nikon Eclipse Ni-E upright microscope equipped with a Hamamastu ORCA-fusionBT Digital CMOS camera. Ciliary beating movies were taken, totaling 1500 frames over 5 s. Kymographs were generated using Fiji: the line tool was used to draw a line parallel to the surface through the cilia of a MCC and the line was resliced through KymoResliceWide plugin.

## Electron microscopy

Embryos at stage 31 were fixed in 0.1 M sodium cacodylate buffer with 4% formaldehyde and 2% glutaraldehyde for 2 h at RT and then overnight at 4 °C. Staining with 1% osmium tetroxide and 0.5% uranyl acetate for 1 h and dehydration with ethanol solutions (35%, 50%, 70%, 95%, and 100%) were followed. The additional dehydration with tetramethylsilane was conducted for samples for scanning EM, followed by air-dry. The dehydrated samples for transmission EM were sunk in Embed 812 resin overnight and then placed in a 55 °C incubator for 48 h. Resin blocks were sectioned by 70–80 nm and stained with 0.5% aqueous uranyl acetate and Reynolds lead citrate. Hitachi S4500 SEM and Hitachi H7600 TEM were utilized to generate images.

## Wholemount in situ hybridization

Embryos were fixed with MEMFA (100 mM MOPS, pH 7.4, 2 mM EGTA, 1 mM MgSO4, and 3.7% formaldehyde) and dehydrated in methanol. Rehydrated embryos with 0.1% Tween20 containing PBS were refixed in 4% paraformaldehyde and incubated with RNA probes in hybridization buffer overnight at 60 °C, followed by washing with 2X SSC and 0.2X SSC solutions. After blocking in MAB buffer with 2% BMB and 10% goat serum, embryos were incubated with the anti-Digoxigennin-AP antibody (11093274910, Roche) in a cold room overnight and washed with MAB buffer. A chromogenic reaction with BM purple substrate (11442074001, Roche) was conducted at room temperature.

## Immunoblot

Using TNSG buffer (Tris-HCl (50 mM, pH7.4), NaCl (150 mM), NP-40 (1%), glycerol (5%), EDTA-free protease inhibitor cocktail (11836170001; Roche), phenylmethylsulfonyl fluoride (1 mM)[60], Na3VO4 (1 mM), and β-glycerophosphate (1 mM)), embryos were lysed at stage 12. Embryo lysates were loaded on 8% SDS-polyacrylamide gel and transferred to PVDF membrane using eBlot™ L1 fast wet transfer system (GenScript). The transferred proteins were incubated in blocking buffer (TBS (pH8) with 0.1% Tween 20 and 5% non-fat dry milk) for 1 h at RT, followed by immunoblotting with HRP-conjugated anti-GFP antibody (dilution 1:2000, MA5-15256-HRP, ThermoFisher) for 1 h at RT or overnight at 4 °C. Pierce™ western blot signal enhancer (21050, ThermoFisher) was used for protein detection.

## Statistics and reproducibility

Data quantification was performed using GraphPad Prism version 10 software. Experimental groups were compared with control by ordinary one-way ANOVA or unpaired two-tailed Student's t-test to determine statistical significance. Unless otherwise noted in the figure legend, all experiments were repeated at least three times.

## Reporting summary

Further information on research design is available in the Nature Portfolio Reporting Summary linked to this article.

## Data availability

Source data for the main figures and supplementary information are provided in the Supplementary Data. All other data are available from the corresponding author on reasonable request.

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

## Acknowledgements

We thank K. Peifley and V. Magidson for technical support with microscopy and all members of the Cancer and Developmental Biology Laboratory for discussion and suggestions. This research was supported in part by the Intramural Research Program of the National Institutes of Health, National Cancer Institute. The content of this publication does not necessarily reflect the views or policies of the U.S. Department of Health and Human Services, nor does mention of trade names, commercial products, or organizations imply endorsement by the U.S. government.

## Author contributions

M. Lee designed and carried out the experiments with the help of C. Carpenter, Y.-S. Hwang, J. Yoon, and Q. Lu. C. Carpenter assisted with EM. C. J. Westlake and S. A. Moody contributed to discussion and interpretation and edited the manuscript. M. Lee and I.O. Daar wrote the manuscript. T.P. Yamaguchi and I.O. Daar supervised the project. All of the authors discussed the results and commented on the manuscript.

## Funding

## Competing interests

The authors declare no competing interests.
