## [Transparent Peer Review file · Communications Biology]

Proliferation associated 2G4 is required for the ciliation of vertebrate motile cilia

Corresponding Author: Dr Ira Daar

Version 0:

Reviewer comments:

Reviewer #1

(Remarks to the Author)

In the manuscript "Proliferation associated 2G4 is required for the ciliation of vertebrate motile cilia" by Lee et al., the authors discover a novel role for Pa2G4 in *Xenopus* motile cilia formation. They analyze localization of Pa2G4 as well as of domain deletion constructs predominantly in multiciliated cells of the embryonic epidermis, and to some extent also in motile monocilia of the left right organizer (GRP). Loss of function studies indicate that Pa2G4 is required for basal body docking, apical actin organization and cilia formation. Furthermore they analyze ciliary beating, which is reduced upon Pa2G4 knockdown. They also use what appear to be GTPase activity sensors and show that Rac1 activity seems impaired upon Pa2G4 knockdown.

This is solid work, with some beautiful imaging and describing a novel role for a protein in cilia formation/function. While they do gain some interesting novel insights, I feel like the mechanistic aspects of the manuscript would need to be expanded for publication at NatComm. and some additional controls, quantifications and explanations would need to be provided. If the authors can address these points, I believe that the paper would be quite interesting for cell and developmental biologists.

Major issues:

- 1.) It remains unclear to me which functional aspects of Pa2G4 cause which phenotype manifestations: On the one hand, we have basal body docking and polarity defects in multiciliated cells, which would be in line with roles in F-actin organization, interactions between basal bodies and apical/subapical F-actin as well as F-actin fibers required for apical transport. However, the co-localization of Pa2G4 and F-actin is not very convincing (it is actually non-overlapping in Figure 1B). On the other hand, Pa2G4 seems to localize to cilia, but it is unclear if to the membrane or axoneme? Also it isn't clear to me if these are two distinct functions (actin and axoneme) or if they are interconnected. In the cilia localization images, it also looks like some of the green signal is continuous between adjacent cilia –an artifact of image post-processing? No explanation is given. This also relates to the finding that Rac1 activity is affected after Pa2G4 knockdown, and Pa2G4 domain functions. Which phenotypic aspects are Rac1 dependent and which domains of Pa2G4 are thought to have which functions? Please add more functional data and more details on the assays, constructs (reporters as well as Pa2G4 deletions) to clarify the points mentioned above.
- 2.) It is also not clear to me if all types of cilia or just motile cilia require Pa2G4 function? Additional data could be provided by analyzing sensory vs motile cilia of the GRP and the neural tube. Also, when the authors want to make a point that this is a finding transferrable to other vertebrates, some data on cells/tissues/embryos from other species should be added.

Minor issues:

- 1.) Given the experienced authors listed on the manuscript, the quality of the language and wording was sometimes a surprise. The first intro paragraph needs heavy rewriting, and a lot of smaller corrections are necessary, e.g. MCCs generate up to 300 cilia, so "several hundred cilia/centrioles" seems a bit overstated; "multicilia" is a ciliate species but used here to describe multiple cilia in MCCs; some aspects of F-actin biology in relation to cilia are introduced, but others equally relevant ones not (e.g. ciliary adhesions, ezrin functions, both required for transport, docking, polarity and ciliation....).
- 2.) Figures: Channels in Fig 1 are in part mis-labeled (green/cyan). Videos are not clearly labeled and I had to play them in image J because of codec issues with standard software. GRP overviews should be shown (especially as sensory cilia on the lateral edges are naturally shorter and centrally positioned). Axoneme TEMs need to be done more extensively as axoneme arrangement/structures tend to get "fuzzy" towards the tips, similarly to what the authors describe as phenotype. In figure 2f, channels are not labeled at all. Please carefully check these things before resubmission.
- 3.) What are r/p/wGBDs, what are they measuring precisely and what types of controls could be added to demonstrate that they work specific in the presented context?

- 4.) Chibby is a transition zone protein rather than distal appendage, or am I confused?
- 5.) What are the known functions of Rac1 in cilia and MCCs?
- 6.) Cilia number quantification in MCCs is really hard to do, especially using SEMs. The numbers of cilia reported per MCC here are ca 50% lower than previous reports (e.g. <https://doi.org/10.1016/j.ydbio.2020.11.011> and <https://doi.org/10.7554/eLife.66076>).
- 7.) To establish a real effect on left-right development, lateral plate mesoderm markers (e.g. *pitx2c*) and *situs* should be analyzed and shown.

Reviewer #2

(Remarks to the Author)

This study by Lee et al. identifies the role of Proliferation associated 2G4 in formation and functioning of cilia in *Xenopus* embryo. The authors report that knockdown of Pa2G4 reduced ciliary length and number in multiciliated cells (MCC). These cells also showed synchronous cilia beating defect, alluding to defective microtubule structure in the axonemes, as illustrated by TEM analysis. Additionally, 2G4 knockdown resulted in disrupted posterior polarization and decreased length in GRP cilia. There was a reduction in activity of Rac1 GTPase observed upon 2G4 knockdown in both GRP cells and MCCs. Based on the abnormal localization of Cep164 (a distal appendage protein) in 2G4 knockdown, it is speculated that Pa2G4 might play a role in the formation of distal appendages. Using deletion constructs, the authors determine that the RNA binding domain (RBD) and C terminus of 2G4 is required for multiciliation.

Taken together, this study has established Pa2G4 as an important protein for ciliogenesis and functioning of cilia in *Xenopus laevis* embryos. There are a few minor issues the authors should address before the paper is accepted for publication:

1. Authors should describe how they quantified the imaging results shown in Fig 3 and Fig 4.
2. The authors describe that 2G4 in GRP is 'extensively expressed' in their results section. Looking at Fig S1A, it appears that the expression of 2G4 in GRP is relatively less as compared to the neighboring tissue. Thus, 'extensive' might not be the accurate description here.
3. In Fig S1C, the authors should provide 'n' numbers for the bar graph and also show representative image for the rescue group.

Reviewer #3

(Remarks to the Author)

In this manuscript, the authors have demonstrated a novel role of Pa2G4 in ciliogenesis during embryonic development. These findings offer valuable insights into genetic diseases associated with motile cilia. This story is very interesting. The manuscript is well written, and the message is clear. However, there are some issues which need to be addressed before publication.

Major points:

1. Determine the inhibitory effect of 2G4MO and the expression levels of 2G4mRNA in Figures 1 and 2 by quantitative real-time polymerase chain reaction (qPCR) or western blotting (WB).
2. Exploration of the mechanism through which PA2G4 modulates the activity of Rac1, perhaps investigation of a potential direct interaction between them.
3. Clarification the location of the magnified area of the zoomed-in images presented in Figure 1A.

Minor point not requiring experiments unless they have data:

As an RNA-binding protein, why does PA2G4 colocalize with AC-tub and exert its effect on ciliogenesis depending on the RBD domain?

Version 1:

Reviewer comments:

Reviewer #1

(Remarks to the Author)

The revised manuscript has addressed all my previous concerns. It can be accepted for publication. Congrats to the authors.

There were some minor things that the authors could modify before publication.

Intro

- Page 3 line 91f: "Furthermore, the unique structure of the motile cilium, comprising nine microtubule doublets with two central microtubules in the axoneme, facilitates directional beating 17,18."

Maybe good to specify that this is in MCCs, in mono-ciliated cells, the motile cilium can have 0, 2 or 4 central microtubules.

Results

- Page 5 line 126-127: "...along with the broad localization of Pa2G4 transcripts in *Xenopus* larval epidermis."
In the two referenced papers there is very little in situ hybridization data and not in earlier stages. Also Pa2G4 transcripts are rather enriched in neural plate/border but less in epidermis. Would be good to rephrase.

- Page 6 line 159f: "In Pa2G4 morphants, not only 159 was the length of GRP cilia significantly shortened, but the cilia in GRP cells also lacked the 160 typical posterior polarization observed in the control morphants (Supplementary Fig. 1c, d)."
Since these aspects were not quantified and statistically tested, I would not use the term significant.

- Supplementary Fig. 1e: Do you meant Anterior-posterior or rather dorso-ventral in the neural tube sections? Please double-check.

- Page 7 line 212f: "In control MCCs, all rGBD-GFP (Rho marker), pGBD-GFP (Rac1 marker) and wGBD-GFP (Cdc42 marker) co-localized with centrin-RFP (basal body marker), suggesting that active Rho, Rac1, and Cdc42 accumulate at the basal body (Fig. 3c,d and Supplementary Fig. 3)."

With the low mag it is hard to make the point of basal body co-localization (or just enrichment near the apical membrane or the cilium). Perhaps magnifications could be shown in Suppl Fig 3, otherwise perhaps rephrase?

Reviewer #2

(Remarks to the Author)

In the revised manuscript, the authors have adequately addressed all my concerns. I am happy to recommend it for publication.

Reviewer #3

(Remarks to the Author)

The authors have done a satisfactory job addressing most of our comments. Pending opinions on the two other referees on the technical issues, I strongly recommend publication.

Dear Reviewers,

We are submitting a revised manuscript (#COMMSBIO-24-3105) entitled “**Proliferation associated 2G4 is required for the ciliation of vertebrate motile cilia**”. We greatly appreciate the suggestions and comments provided by the reviewers, and we have performed several new experiments to address the majority of concerns raised. In this revised paper, we believe our combination of loss-of-function (endogenous) experiments, replacement experiments (knockdown of endogenous protein followed by re-expression at carefully titrated levels), and *in vivo* assays in *Xenopus* embryos provide clear insights into the role of Pa2G4 in ciliogenesis in multiciliated cells (MCCs) and the gastrocoel roof plate (GRP) and its coordination with Rac1. We are grateful to the reviewers for their valuable suggestions, which have led to a more comprehensive assessment of Pa2G4's role in the ciliation of vertebrate motile cilia, thus strengthening the claims in our paper.

The concerns of reviewers have been addressed below:

Reviewer #1 (Remarks to the Author):

In the manuscript “Proliferation associated 2G4 is required for the ciliation of vertebrate motile cilia” by Lee et al., the authors discover a novel role for Pa2G4 in *Xenopus* motile cilia formation. They analyze localization of Pa2G4 as well as of domain deletion constructs predominantly in multiciliated cells of the embryonic epidermis, and to some extent also in motile monocilia of the left right organizer (GRP). Loss of function studies indicate that Pa2G4 is required for basal body docking, apical actin organization and cilia formation. Furthermore they analyze ciliary beating, which is reduced upon Pa2G4 knockdown. They also use what appear to be GTPase activity sensors and show that Rac1 activity seems impaired upon Pa2G4 knockdown.

This is solid work, with some beautiful imaging and describing a novel role for a protein in cilia formation/function. While they do gain some interesting novel insights, I feel like the mechanistic aspects of the manuscript would need to be expanded for publication at NatComm. and some additional controls, quantifications and explanations would need to be provided. If the authors can address these points, I believe that the paper would be quite interesting for cell and developmental biologists.

We thank the reviewer for their remarks regarding the quality and solid nature of the work in our study for the journal *Communications Biology*. We have addressed the comments below.

Major issues:

1.) It remains unclear to me which functional aspects of Pa2G4 cause which phenotype manifestations: On the one hand, we have basal body docking and polarity defects in

multiciliated cells, which would be in line with roles in F-actin organization, interactions between basal bodies and apical/subapical F-actin as well as F-actin fibers required for apical transport. However, the co-localization of Pa2G4 and F-actin is not very convincing (it is actually non-overlapping in Figure 1B).

We agree with the reviewer that it is difficult to assign specific functions to specific phenotypes, but several defects will align with actin meshwork disruption.

We also agree with the reviewer's comment that GFP-Pa2G4 does not completely overlap with F-actin. To clarify the Pa2G4 localization data, we have revised the sentence in the Result section. The corrected sentence on lines 9-12 of page5 is as follows:

Interestingly, we observed strong GFP-Pa2G4 signals on the surface of MCCs. GFP-Pa2G4 partially overlapped with or was juxtaposed to the phalloidin-stained apical actin meshwork and was largely excluded from the distal appendage area, marked by Cep164 staining.

On the other hand, Pa2G4 seems to localize to cilia, but it is unclear if to the membrane or axoneme?

To address the reviewer's question, we tested GFP-Pa2G4 localization by co-expressing with membrane-RFP (a membrane marker) and acetylated tubulin staining (an axoneme marker) by 3D-Structured Illumination Microscopy (SIM). SIM showed GFP-Pa2G4 likely localizes to both membrane-RFP and acetylated tubulin (Fig.1a). However, since the membrane-RFP and acetylated tubulin signals from SIM were not fully distinguishable from each other, the exact localization of Pa2G4 to the ciliary membrane, axoneme or both remains unclear.

Also it isn't clear to me if these are two distinct functions (actin and axoneme) or if they are interconnected.

We believe our data supports the concept that actin network formation, basal body docking, ciliary growth and function (beating) are interconnected; if the actin network is deregulated in MCCs, basal bodies fail to dock to the actin meshwork, compromising ciliary growth (axoneme growth). Conversely, ciliary movement is required for actin network formation in MCCs (Mahuzier et al., 2018), leading to synchronized ciliary beating. Since our data showed Pa2G4 localization to both cilia and the apical actin meshwork, Pa2G4 function at the apical surface of MCCs could affect ciliary formation and movement by contributing to basal body docking. Although the exact role of ciliary Pa2G4 remains unclear in our study, we observed Pa2G4 knockdown caused the central pair to be missing in MCC cilia. Given that abnormal axonemal structures impair synchronized cilia beating in MCCs and the cilia beating is required for actin

network formation, the roles of Pa2G4 in actin meshwork and cilia (axoneme) are likely interconnected.

In the cilia localization images, it also looks like some of the green signal is continuous between adjacent cilia –an artifact of image post-processing? No explanation is given.

As the reviewer points out, some of the green signals appear continuous between adjacent cilia. These signals are not an artifact of image post-processing. While acetylated tubulin signals are typically strong, their intensity can vary among cilia. To avoid signal saturation, the acetylated tubulin signals were adjusted broadly across the image, which caused some cilia signals to appear too weak, giving the impression that cilia were absent. However, cilia are present, and the green signals are visible on the cilia.

We understand the reviewer's concern and have replaced the previous images with new ones (Fig. 1a) that now include the membrane-RFP signal for clarity.

This also relates to the finding that Rac1 activity is affected after Pa2G4 knockdown, and Pa2G4 domain functions. Which phenotypic aspects are Rac1 dependent and which domains of Pa2G4 are thought to have which functions? Please add more functional data and more details on the assays, constructs (reporters as well as Pa2G4 deletions) to clarify the points mentioned above.

To address the reviewer's question, we first tested which domains of Pa2G4 are involved in basal body migration and docking. Consistent with Figure 5, which shows that the RBD and C-terminal region of Pa2G4 are required for actin meshwork formation and cilia growth in MCCs, both the Δ RBD and Δ C constructs failed to rescue impaired basal body docking. This suggests that the RBD and C-terminal regions play an important role in basal body docking and subsequent cilia growth.

We further tested which regions of Pa2G4 are required for regulating Rac1 activity. Interestingly, while the expression of Pa2G4 WT and Δ N rescued the decreased Rac1 activity, Δ RBD and Δ C expression did not restore the Rac1 activity in Pa2G4 morphant MCCs, indicating that the RBD and C-terminal regions may be required for Rac1 regulation in MCCs. Thus, Pa2G4 may coordinate with Rac1 to regulate ciliogenesis, including basal body docking, actin network formation, ciliary formation, and movement.

These data are now included in Figure 5.

2.) It is also not clear to me if all types of cilia or just motile cilia require Pa2G4 function? Additional data could be provided by analyzing sensory vs motile cilia of the GRP and the neural tube. Also, when the authors want to make a point that this is a finding transferrable to other vertebrates, some data on cells/tissues/embryos from other species should be added.

The reviewer poses an interesting question of whether only motile cilia are affected.

To test if Pa2G4 is required for primary ciliogenesis in the neural tube, Pa2G4 morpholinos were injected into one dorsal marginal zone of 8 cell stage embryos to target the neural tube, and the embryos at stage 25 were transversely sectioned, followed by staining with acetylated tubulin. Interestingly, compared to the control side (average cilia population per section: 33), the Pa2G4 MO-injected side of the neural tube contained a reduced cilia population (average cilia population per section: 12), suggesting that Pa2G4 also plays a role in primary ciliogenesis in the neural tube. We now include this data in Supplementary Figure 1e.

While we also examined the sensory cilia of the GRP, we did not observe dramatic changes in sensory cilia length and number in Pa2G4 morpholino injected side relative to the control side. It is possible that this analysis is somewhat confounded by the naturally shorter length of sensory cilia found in the GRP.

Although we agree that it would be interesting to expand our studies into mammals, we believe it is important to perform our experiments *in vivo*, in the normal context of a vertebrate. Thus, mammalian studies are beyond the scope of the current study.

Minor issues:

1.) Given the experienced authors listed on the manuscript, the quality of the language and wording was sometimes a surprise. The first intro paragraph needs heavy rewriting, and a lot of smaller corrections are necessary, e.g. MCCs generate up to 300 cilia, so “several hundred cilia/centrioles” seems a bit overstated; “multicilia” is a ciliate species but used here to describe multiple cilia in MCCs; some aspects of F-actin biology in relation to cilia are introduced, but others equally relevant ones not (e.g. ciliary adhesions, ezrin functions, both required for transport, docking, polarity and ciliation....).

In response to the reviewer's suggestions, we revised the introduction by adding more information and correcting our word usage. The changes in the introduction are as follows:

**line 5, page 3 : up to three hundred cilia
multicilia to multiple cilia**

line 18-31, page 3 : we added more information as indicated by the reviewer

2.) Figures: Channels in Fig 1 are in part mis-labeled (green/cyan). Videos are not clearly labeled and I had to play them in image J because of codec issues with standard software.

We corrected the labeling in Fig. 1, the figure legend, and the video files. Regarding playback with standard software, we are sorry for any inconvenience and we have tested the videos using Fiji, VLC media player, and MAC QuickTime player, and the video files play well in all of these programs.

GRP overviews should be shown (especially as sensory cilia on the lateral edges are naturally shorter and centrally positioned).

The GRP overview images are now presented in Supplementary Figure 1d.

Axoneme TEMs need to be done more extensively as axoneme arrangement/structures tend to get “fuzzy” towards the tips, similarly to what the authors describe as phenotype. In figure 2f, channels are not labeled at all. Please carefully check these things before resubmission.

As the reviewer mentioned, the axoneme arrangement and structures are likely to get fuzzy towards the tips. However, the most distal tip of motile cilia consists of central microtubules and peripheral singlets with significantly narrowed diameters (Osinka et al., 2019). Compared to control TEM images, the sectioned TEM images from Pa2G4 knockdown MCCs show similar diameters. In addition, the Pa2G4 morphant TEM images contain outer doublet microtubules, like control TEMs. The defective ciliary structure is most likely due to Pa2G4 knockdown, rather than being a TEM artifact.

Regarding the channel labeling in Figure 2f, the channel information has now been added to the figure.

3.) What are r/p/wGBDs, what are they measuring precisely and what types of controls could be added to demonstrate that they work specific in the presented context?

Rho family GTPases (e.g. RhoA, Rac1 and Cdc42) affect cytoskeletal dynamics only in their active (GTP-bound) conformation. Fluorescent protein-tagged rGBD, pGBD and wGBD can be used to monitor the spatiotemporal activity of these GTPases. The GTPase binding domains (GBDs) of effector proteins bind only to the active conformation of Rho GTPases. Thus, active Rho, Rac and Cdc42 are detected by rGBD (GBD of Rhotekin), pGBD (GBD of PAK3), and wGBD (GBD of N-WASP), respectively. Fluorescent protein-tagged GBD probes have been widely used to study Rho GTPases dynamics during *Xenopus* development.

As for controls, GFP alone could serve as a control. However, in our experiments, using GFP-rGBD and GFP-wGBD are more specific controls because only GFP-pGBD signal responded to Pa2G4 knockdown in MCCs, while GFP-rGBD and GFP-wGBD did not. To validate that the loss of GFP-pGBD signal from basal bodies was not due to mislocalization of the Rac1 protein in the Pa2G4 morphant

MCCs, we showed that exogenously expressed Rac1-GFP was unaltered in the morphants.

We now describe this in more detail for clarity in the manuscript on page 7 (lines 21-25).

4.) Chibby is a transition zone protein rather than distal appendage, or am I confused?

In airway multiciliated epithelial cells, Cby is recruited to the distal appendages of centrioles via its physical interaction with the distal appendage protein CEP164. Cby then associates with Rabin8, a component of the membrane trafficking machinery and a guanine nucleotide exchange factor for the small guanosine triphosphatase Rab8, to facilitate the recruitment of Rab8 and promote efficient assembly of ciliary vesicles (Burke et al, 2014). As the reviewer noted, however, *Drosophila* Cby associates with the basal body transition zone in sensory neurons and male germ cells in *Drosophila* (Enjolras et al, 2012).

5.) What are the known functions of Rac1 in cilia and MCCs?

Rac1 plays a role of ciliogenesis and cilia function. Rac1 increases IFT88 stability and reduced IFT88 levels compromise ciliogenesis in the C3H10T1/2 cell line (Tang et al., 2022). Also, Rac1 is required for the polarized localization of the basal body in the node cells of mouse embryos (Hashimoto et al., 2010). Defects in Rac1 activity are associated with abnormal primary cilia assembly in Lowe syndrome (Madhivanan et al., 2012). In zebrafish, Rac1 knockdown causes shortened cilia in the pronephric tubule and defects in cardiac looping. During *Xenopus* epidermis development, Rac1 knockdown results in irregular basal body docking and spacing in MCCs (Epting et al., 2015).

We now convey this information in the discussion of the manuscript (lines 23-29, page 9).

6.) Cilia number quantification in MCCs is really hard to do, especially using SEMs. The numbers of cilia reported per MCC here are ca 50% lower than previous reports (e.g. <https://doi.org/10.1016/j.ydbio.2020.11.011> and <https://doi.org/10.7554/eLife.66076>).

As the reviewer suggested, we quantified cilia number by counting the apical centrioles in MCCs. In counting the centrioles from more than 100 MCCs, the average centriole number in control morphant MCCs was 137, which falls within the range reported in previous studies. The average centriole number in 2G4 morphant and rescued MCCs was 106 and 121, respectively. The reduction in centriole number per MCC upon 2G4 knockdown (conMO:2G4MO:rescue=1:0.77:0.88) was similar to the reduction in cilia number observed in SEM data (conMO:2G4MO:rescue=1:0.75:0.89).

However, we have replaced the previous data with this new quantification (Figure 2c).

7.) To establish a real effect on left-right development, lateral plate mesoderm markers (e.g. *pitx2c*) and situs should be analyzed and shown.

As the reviewer suggested, we performed whole-mount in situ hybridization using the *pitx2c* probe. Control morphants (50/52) showed wild-type expression in the left-lateral plate mesoderm. However, depending on MO concentration, most 2G4 morphants lacked *pitx2c* expression (2G4MO(3ng); 30/61, 2G4MO(5ng); 50/58) and only a small portion of 2G4 morphants showed bilateral expression (2G4MO(3ng); 5/61, 2G4MO(5ng); 2/58), suggesting that Pa2G4 knockdown causes laterality defects.

This data is now displayed in supplementary Figure 2b.

Regarding situs inversion, we did not observe a clear situs inversion phenotype but rather heterotaxia in 2G4 morphants (please see below, this data is not included in the manuscript). The embryos injected into the dorsal marginal zone to target GRP cilia, exhibited a short and bent embryonic axis, small anterior structures, and defective organogenesis. Although the knockdown phenotypes could be partially explained by the role of Pa2G4 in neural crest (Karen Neilson, 2017), they may also result from an unknown developmental role of Pa2G4. Thus, evaluating GRP cilia phenotypes at later stages may prove challenging and may be obfuscated by secondary effects.

Reviewer #2 (Remarks to the Author):

This study by Lee et al. identifies the role of Proliferation associated 2G4 in formation and functioning of cilia in Xenopus embryo. The authors report that knockdown of Pa2G4 reduced ciliary length and number in multiciliated cells (MCC). These cells also showed synchronous cilia beating defect, alluding to defective microtubule structure in the axonemes, as illustrated by TEM analysis. Additionally, 2G4 knockdown resulted in disrupted posterior polarization and decreased length in GRP cilia. There was a

reduction in activity of Rac1 GTPase observed upon 2G4 knockdown in both GRP cells and MCCs. Based on the abnormal localization of Cep164 (a distal appendage protein) in 2G4 knockdown, it is speculated that Pa2G4 might play a role in the formation of distal appendages. Using deletion constructs, the authors determine that the RNA binding domain (RBD) and C terminus of 2G4 is required for multiciliation.

Taken together, this study has established Pa2G4 as an important protein for ciliogenesis and functioning of cilia in *Xenopus laevis* embryos. There are a few minor issues the authors should address before the paper is accepted for publication:

We appreciate the reviewer's assessment of our study and we address the specific comments below.

1. Authors should describe how they quantified the imaging results shown in Fig 3 and Fig 4.

As the reviewer suggested, we included quantification graphs in Figures 3 and 4, along with detailed quantification information in the figure legends.

2. The authors describe that 2G4 in GRP is 'extensively expressed' in their results section. Looking at Fig S1A, it appears that the expression of 2G4 in GRP is relatively less as compared to the neighboring tissue. Thus, 'extensive' might not be the accurate description here.

We agree with the reviewer that more precise language is required. We edited the description as follows :

line 3, page 6 : Pa2G4 is widely expressed in the roof of the gastrocoel.

3. In Fig S1C, the authors should provide 'n' numbers for the bar graph and also show representative image for the rescue group.

We now include 'n' numbers for the bar graph and a representative image for the rescue group.

Reviewer #3 (Remarks to the Author):

In this manuscript, the authors have demonstrated a novel role of Pa2G4 in ciliogenesis during embryonic development. These findings offer valuable insights into genetic diseases associated with motile cilia. This story is very interesting. The manuscript is well written, and the message is clear. However, there are some issues which need to be addressed before publication.

We appreciate the reviewer's remarks regarding the value of our study.

Major points:

1. Determine the inhibitory effect of 2G4MO and the expression levels of 2G4mRNA in Figures 1 and 2 by quantitative real-time polymerase chain reaction (qPCR) or western blotting (WB).

Since a commercial antibody to detect endogenous *Xenopus* Pa2G4 protein is not available, we tested the translation inhibitory effect of the 2G4MO using exogenously expressed HA-tagged Pa2G4. WB analysis showed the 2G4 MO specifically decreased wild-type HA-Pa2G4 expression, but the MO-resistant (with 6 nucleotide mutations in the MO binding region) construct remained unaffected by 2G4 MO, confirming the specificity of 2G4 MO. We now include the data in Supplementary Figure 1.

Regarding the expression levels of 2G4 mRNA in Figures 1 and 2, both pa2g4.L and pa2g4.S mRNA seq data is publicly available and show that both S and L forms are highly expressed in tadpoles at stages 25-30.

(<https://www.xenbase.org/xenbase/gene/geneExpressionChart.do?method=draw&geneId=943132&geneSymbol=pa2g4>).

Also, whole-mount in situ hybridization demonstrated a broad spatial expression of Pa2G4 on the epidermis of tadpoles (Neilson et al., 2010 & 2017), thus the epidermis (including MCCs) of embryos used in Figures 1 and 2 should be expressing substantive levels of Pa2G4.

2.Exploration of the mechanism through which PA2G4 modulates the activity of Rac1, perhaps investigation of a potential direct interaction between them.

As the reviewer suggested, we conducted co-IP to test the interaction between Pa2G4 and Rac1. We co-expressed Flag-Pa2G4 with HA-Rac1 or HA-Six1 and immunoprecipitated the proteins using either Flag or HA beads. While Six1, a known binding partner of Pa2G4, interacted with Pa2G4, Rac1 did not (please see below, this data is not included in the manuscript).

We then assessed what domain of Pa2G4 is required for Rac1 activity. Although the expression of Pa2G4 WT and ΔN rescued the decreased Rac1 activity, the expression of ΔRBD and ΔC failed to recover Rac1 activity in Pa2G4 morphant MCCs. This suggests that the RBD and C-terminal regions may be critical for Rac1 activation in MCCs. Thus, the coordination of Pa2G4 and Rac1 may play a role in basal body docking, actin network formation and ciliary formation and function.

We now include the data in Figure 5.

3. Clarification the location of the magnified area of the zoomed-in images presented in Figure 1A.

We replaced the previous figure with a new one in Figure 1a for clarity.

Minor point not requiring experiments unless they have data:

As an RNA-binding protein, why does PA2G4 colocalize with AC-tub and exert its effect on ciliogenesis depending on the RBD domain?

While we have no data to speak to this question, we can speculate. Multiple cilia of mouse ependymal cells are abundant in RNAs including rRNA and mRNA, and mRNA binding protein Fmrp localizes in cilia and functions in mRNA delivery into the cilia during ciliogenesis (Kai Hao et al., 2021). Although it is not clear if Pa2G4 directly delivers RNAs into the cilia, Pa2G4 as a binding partner of ribosomal proteins (Varun Bhaskar et al., 2021), might localize in cilia with other protein-RNA complexes to regulate ciliogenesis.

Another possibility is that the RBD domain of Pa2G4 may interact with potentially important protein binding partners, rather than RNA, in a context dependent manner. We observed that Pa2G4 ΔRBD failed to rescue Rac1 activity in MCCs

(Figure 5). Although it is beyond the scope of this study, perhaps Pa2G4 interacts with an upstream regulator of Rac1 (e.g. Vav1 and Dock).

REVIEWERS' COMMENTS:

Reviewer #1 (Remarks to the Author):

The revised manuscript has addressed all my previous concerns. It can be accepted for publication. Congrats to the authors.

There were some minor things that the authors could modify before publication.

Intro

- Page 3 line 91f: "Furthermore, the unique structure of the motile cilium, comprising nine microtubule doublets with two central microtubules in the axoneme, facilitates directional beating^{17,18}."

Maybe good to specify that this is in MCCs, in mono-ciliated cells, the motile cilium can have 0, 2 or 4 central microtubules.

We included "in MCCs" in Page 3, line 30:

Furthermore, the unique structure of the motile cilium **in MCCs**, comprising nine microtubule doublets with two central microtubules in the axoneme, facilitates directional beating^{17,18}

Results

- Page 5 line 126-127: "...along with the broad localization of Pa2G4 transcripts in *Xenopus* larval epidermis."

In the two referenced papers there is very little in situ hybridization data and not in earlier stages. Also Pa2G4 transcripts are rather enriched in neural plate/border but less in epidermis. Would be good to rephrase.

We partially agree with the reviewer's comment. However, the reference paper (Neilson, 2010, Figure 9) clearly showed the localization of Pa2G4 transcripts in *Xenopus* larval epidermis at neurula and tailbud stages. Although the intensity of Pa2G4 transcripts in epidermis is relatively weak compared to the cranial neural crest and brain regions at the tailbud stage, the transversely sections display weak but broad expression of Pa2G4 in the epidermis.

However, to avoid giving the impression of exclusive expression of Pa2G4 in the epidermis, we removed the word 'broad' from the sentence:

Page 5, line 4-5: ...along with the localization of Pa2G4 transcripts in *Xenopus* larval epidermis.

- Page 6 line 159f: "In Pa2G4 morphants, not only 159 was the length of GRP cilia significantly shortened, but the cilia in GRP cells also lacked the 160 typical posterior

polarization observed in the control morphants (Supplementary Fig. 1c, d).”
Since these aspects were not quantified and statistically tested, I would not use the term significant.

We removed the term ‘significant’ from the manuscript.

- Supplementary Fig. 1e: Do you meant Anterior-posterior or rather dorso-ventral in the neural tube sections? Please double-check.

We have corrected from (anterior and posterior) to (dorsal and ventral) in supplementary Fig. 1e.

- Page 7 line 212f: “In control MCCs, all rGBD-GFP (Rho marker), pGBD-GFP (Rac1 marker) and wGBD-GFP (Cdc42 marker) co-localized with centrin-RFP (basal body marker), suggesting that active Rho, Rac1, and Cdc42 accumulate at the basal body (Fig. 3c,d and Supplementary Fig. 3).”

With the low mag it is hard to make the point of basal body co-localization (or just enrichment near the apical membrane or the cilium). Perhaps magnifications could be shown in Suppl Fig 3, otherwise perhaps rephrase?

We have rephrased the sentence as follows:

Page 7, line 31-Page 8, line 1-3:

In control MCCs, all rGBD-GFP (Rho marker), pGBD-GFP (Rac1 marker) and wGBD-GFP (Cdc42 marker) were enriched near centrin-RFP (basal body marker), suggesting that active Rho, Rac1, and Cdc42 accumulate in the basal body area (Fig. 3c,d and Supplementary Fig. 3).

Reviewer #2 (Remarks to the Author):

In the revised manuscript, the authors have adequately addressed all my concerns. I am happy to recommend it for publication.

Reviewer #3 (Remarks to the Author):

The authors have done a satisfactory job addressing most of our comments. Pending opinions on the two other referees on the technical issues, I strongly recommend publication.

Supplementary Movies:

File name: Supplementary Movie 1

Description: Multi-cilia beating in representative control morphant. Frame rate 300 frames/second.

File name: Supplementary Movie 2

Description: Multi-cilia beating in representative Pa2G4 morphant. Frame rate 300 frames/second.

File name: Supplementary Movie 3

Description: Multi-cilia beating in representative Rescue embryo. Frame rate 300 frames/second.